# Somatostatin receptors in fibrotic myocardium

**Estibaliz Castillero**[1,2], **Chiara Camillo**[1], **W. Clinton Erwin**[1], **Sameer Singh**[1], **Nafisa Mohamoud**[1], **Isaac George**[1], **Elizabeth Eapen**[3], **Keith Dockery**[3], **Giovanni Ferrari**[1,4], **Himanshu Gupta**[5]*

**1** Department of Surgery, Columbia University; New York, NY, United States of America, **2** Department of Physiology, University of the Basque Country (UPV/EHU), Leioa, Spain, **3** Valley Health System; Ridgewood, NJ, United States of America, **4** Department of Biomedical Engineering, Columbia University, New York, NY, United States of America, **5** Valley Hospital Heart and Vascular Institute; Ridgewood, NJ, United States of America

* GUPTHI@Valleyhealth.com

## Abstract

A patient with a neuroendocrine tumor and history of coronary artery disease underwent PET with 68Ga-DOTATATE PET tracer for tumor visualization. Analysis of the scan showed uptake of 68Ga-DOTATATE in the left ventricle corresponding to previous myocardial infarct. 68Ga-DOTATATE binds by somatostatin receptors (SSTR) and it has been proposed that it may be useful for the detection of cardiac inflammatory lesions. We aimed to test whether SSTR could be upregulated in cardiac fibrotic scar. We analyzed SSTR in cardiac samples from patients with end-stage ischemic cardiomyopathy (ICM, n = 8) and control hearts (n = 5). In mature ICM tissue, *SSTR1* and *SSTR2* expression was unchanged and *SSTR5* expression was significantly decreased in ICM samples vs. control. Immunohistochemistry showed increased SSTR1 and SSTR2 in ICM. Areas with SSTR1 or SSTR2 staining were often adjacent to fibrotic areas. The majority of SSTR1 and SSTR2 staining localized in cardiomyocytes in fibrotic scar-rich areas where CD68 macrophage staining was not present. SSTR are occasionally upregulated in cardiac fibrotic areas. When using 68Ga-DOTATATE PET tracer to detect cardiac sarcoidosis or atherosclerotic plaque, the possibility of tracer uptake in fibrotic areas should be considered.

## Background

A patient with a neuroendocrine tumor and previous history of coronary artery disease underwent PET with 68Ga-DOTATATE PET tracer for tumor visualization. The study demonstrated 68Ga-DOTATATE uptake not only in the tumor but also in the left ventricle (LV) myocardium corresponding to previous myocardial infarct. 68Ga-DOTATATE binds by somatostatin receptors (SSTR), making it useful for the imaging of tumors that have a high abundance of SSTR, such as neuroendocrine tumors. 68Ga-DOTATATE has the highest affinity for SSTR subtype 2, with lower affinity for SSTR1 and SSTR3 [1]. Uptake of 68Ga-DOTATATE has been reported, and is expected, in tumor-free tissues including pituitary gland,

**Data Availability Statement:** All relevant data are within the manuscript and its Supporting Information files.

**Funding:** This work was supported, in part, by a National Heart, Lung and Blood Institute of the

National Institutes of Health (R01-HL131872) (to G.F.). S.S. received funding from the Thoracic Surgery Foundation Resident Research Award. The funders (National Heart, Lung and Blood Institute of the National Institutes of Health and Thoracic Surgery Foundation) had no role in study design, data collection and analysis, decision to publish, or preparation of the manuscript

**Competing interests:** The authors have declared that no competing interests exist.

spleen, liver, adrenal glands, and urinary tract [2, 3]. However, no cardiac uptake of 68Ga-DOTATATE is expected in healthy hearts. Activated inflammatory cells, including epithelioid cells, multinucleated giant cells and some macrophages have abundant SSTR presence [4]. A previous study [5] investigated the usefulness of 68Ga-DOTATATE for the detection of sarcoidosis taking advantage of the low signal of the tracer in the heart at baseline. Despite the low sensitivity of 68Ga-DOTATATE to detect cardiac inflammation, there was increased uptake in hearts with sarcoidosis. Further, on immunohistochemistry staining, SSTR2 localized in granuloma-rich myocardial areas of hearts with sarcoidosis. Due to the ability of 68Ga-DOTATATE to detect activated inflammatory cells, clinical studies and animal experiments have explored its utility to identify inflammation in several cardiovascular models. 68Ga-DOTATATE aided in visualizing atherosclerotic plaque in mice [6], and this approach was shown to be feasible in patients [7]. 68Ga-DOTATATE was useful to detect lymphocyte infiltration in acute allograft rejection in cardiac transplantation in rats [8], moreover, it was reported that 68Ga-DOTATATE could non-specifically detect necrosis and infarct due to its signal in inflammatory cells. However, the relationship of SSTR expression in myocardial scar is not well described. We therefore evaluated the SSTR expression and its relationship to inflammatory macrophages in end-stage ischemic cardiomyopathy (ICM).

## Methods

### Human specimens

This study met all institutional guidelines of the Institutional Review Board of Columbia University, New York State organ donation guidelines and The Valley Hospital regarding informed consent, the use of clinical data, ethical treatment of patients by adhering to the principles of the Declaration of Helsinki, and procurement of tissue for research. All subjects were recruited at the New York Presbyterian Hospital-Columbia University campus between 2010 and 2021.

**Control patients.** Subjects with no known cardiopulmonary disease whose organs were listed but were unable to be placed at the time of organ recovery for heart transplantation and who consented to donate tissue for research purposes by Live On New York (previously New York Organ Donor Network) were included in this study (n = 5). These hearts ended up not being used for logistic reasons or size, not due to any problems with the heart. At the time of heart reception, a specimen was obtained from the apex of the LV. Specimens were subdivided and a portion of each sample was immediately flash frozen in liquid nitrogen and stored at −80˚C, while a portion was stored in formalin for histology analysis.

**Patients with end-stage ischemic cardiomyopathy.** Patients with end-stage dilated cardiomyopathy from ischemic origin with myocardial tissue previously collected by Columbia Department of Surgery biobanks were included in this study (n = 8). The cohort included patients who underwent heart transplantation (n = 2) and patients who underwent left ventricle assist device (LVAD) implantation at the New York Presbyterian-Columbia campus (n = 6). Patients in cardiogenic shock at the time of LVAD implantation, possible viral etiology, surgical complications after LVAD, chronic infection, or right ventricular failure were excluded from the analysis. Ischemic origin was identified by clinical data review, noting previous percutaneous coronary intervention, coronary artery bypass grafting and/or myocardial infarction. At the time of heart transplantation or LVAD implantation a specimen was obtained from the apex of the LV. Specimens were subdivided and a portion of each sample was immediately flash frozen in liquid nitrogen and stored at −80˚C, while a portion was stored in formalin for histology analysis. Demographic and clinical data for subjects were collected via chart review.

### RNA isolation and gene expression analysis

Isolation of total mRNA for real-time PCR was performed using the RNeasy mini kit (Qiagen, Valencia, CA). 30 mg of flash-frozen frozen tissue were homogenized with a TissueRuptor (Qiagen). RNA concentration was measured on a DS11 spectrophotometer (DeNovix, Wilmington, DE). For PCR, 100ng of RNA were retro-transcribed with a Maxima H Minus cDNA Synthesis Master Mix kit (Thermo Scientific, Whaltham, MA). Gene expression was measured for SSTR with Thermo Taqman probes in a QuantStudio™ 5 Real-Time PCR System (Applied Biosystems, Thermo). *GAPDH* was used as a housekeeping gene. To assess the status of active cardiac remodeling in the cardiac specimens included in the study, we measured gene expression of markers of cardiac stress (brain natriuretic peptide, *BNP*, and myostatin, *MSTN*) and fibrotic remodeling (alpha smooth muscle actin, *ACTA2*, collagen-1, *COL1A1* and transforming growth factor beta-1, *TGFβ1*). The Taqman probes used were: *SSTR1*, Hs00265617_s1; *SSTR2*, Hs00265624_s1; *SSTR5*, Hs00990407_s1; *GAPDH*, Hs99999905_m1; *TGFβ1*, Hs00998133_m1. For the remaining genes, Sybr green based primers were used with the following sequences: *BNP* forward, 5′- cgcaaaatggtcctctacac-3′; *BNP* reverse, 5′- ccgtggaaattttgtgctc-3′; *MSTN* forward, 5′-tggtcatgatcttgctgtaacc-3′; *MSTN* reverse, 5′-cttgacctctaaaaacggattca-3′; *COL1A1* forward, 5′-gggattccctggacctaaag-3′; *COL1A1* reverse, 5′-ggaacacctcgctctcca-3′; *ACTA2* forward, 5′-cctatccccgggactaagac-3′; *ACTA2* reverse, 5′-aggcagtgctgtcctctttct-3′. Results were calculated with the 2-ΔΔCT method and expressed as a percentage of expression in control samples.

### Histology and immunohistochemistry analysis

Standard histology was performed in 5μm sections with the following stains/antibodies: SSTR1, #PA3-108, Invitrogen, polyclonal in rabbit, 1:2500 dilution; SSTR2, #PA3-109, Invitrogen, polyclonal in rabbit, 1:2500 dilution; SSTR1, Abcam AB137083 monoclonal in rabbit, 1:500 dilution; SSTR2, Abcam AB134152 monoclonal in rabbit, 1:1000 dilution; SSTR5, #PA3-112, Invitrogen, polyclonal in rabbit 1:2500; CD68 #ab213363, Abcam, monoclonal in rabbit 1:8000 dilution; Picro-Sirius red stain, Abcam. Briefly, sections of paraffin-embedded tissues were deparaffined by 30 min incubation at 70°C and three washes of 100% xylene. Samples were rehydrated in progressively diluted ethanol and subsequently underwent antigen retrieval by incubation in citrate buffer (Sigma) at 60°C. Samples underwent blocking in 3% bovine serum albumin diluted in phosphate-buffer saline. Incubation with primary antibodies was conducted in a humidity chamber overnight at 4°C with the above indicated concentrations diluted in Dako wash buffer (Dako). Quenching of the endogenous peroxidase was performed by incubation of samples for 10 min with 3% $H_2O_2$. After 1h incubation with HPR-linked secondary antibodies (Abcam), development was performed by DAB Substrate Kit (Abcam) with subsequent hematoxylin counterstain (Sigma). Stained samples were dehydrated and cleared and mounted with Permount mounting medium (Sigma).

### Data analysis

Gene expression results are reported as means ± standard error. No data were excluded from the analysis. Statistical analysis was performed by Student t test (ICM versus control). Pearson's product-moment correlation coefficient was used to determine the correlation between gene expression markers. $P<0.05$ was considered significant. Data were analyzed with the Prism software (GraphPad Software, San Diego, CA).

## Results

### Gene expression of SSTR in hearts with ICM

Table 1 shows demographic and clinical data of the ICM patients included in the study. All patients were maintained on a heart failure medication regimen. None of the patients were treated with SSTR agonists.

Cardiac expression of *SSTR* had been previously reported to be cell-specific, with cardiomyocytes reported to express only *SSTR1* and *SSTR2* and fibroblasts expressing *SSTR1*, *SSTR2*, *SSTR4* and *SSTR5* [9]. We conducted a preliminary screen of previous RNAseq counts [10] in control hearts and hearts of patients with dilated cardiomyopathy (from both ischemic and non-ischemic origin). None of the *SSTR* sequences were significantly changed in patients with dilated cardiomyopathy in comparison with control hearts. A "fragments per kilobase per million mapped fragments (FPKM)" value equal to 0.1 is often considered the minimum FPKM

**Table 1. Demographic and clinical data of ICM patients.** MI, myocardial infarction; PCI, percutaneous coronary intervention; CABG, coronary artery bypass grafting; HF, heart failure; LVEF, left ventricular ejection fraction; LVEDD, left ventricular end-diastolic diameter; IVS, interventricular septum; BNP, brain natriuretic peptide; ACE, angiotensin-converting enzyme; ARB, angiotensin-II receptor blockers.

|  | ICM patients (N = 8) |
| --- | --- |
| Age, median (range) | 64.5(53–75) |
| Gender (male), N(%) | 4(50) |
| Weight (kg) | 78(60–100) |
| Comorbidities |  |
| Hypertension | 5(62.5) |
| Diabetes | 6(75) |
| Hyperlipidemia | 3(37.5) |
| Chronic Kidney Disease | 3(37.5) |
| Atrial Fibrillation | 2(25) |
| Prior MI | 5(62.5) |
| Prior PCI | 4(50) |
| Prior CABG | 5(62.5) |
| Duration HF (years) | 13.5(2–20) |
| Preoperative Echo |  |
| LVEF (%) | 10(10–25) |
| LVEDD (cm) | 7.4(6.0–8.5) |
| IVS, thickness (cm) | 0.9(0.6–1.1) |
| Preoperative Labs |  |
| Creatinine (mg/dL) | 1.16(1.14–1.72) |
| BNP (pg/mL) | 885(119–2703) |
| Preoperative Medications |  |
| Somatostatin analogues | 0(0) |
| Aspirin | 8(100) |
| β-Adrenergic Antagonists | 8(100) |
| ACE inhibitors | 2(25) |
| Statins | 6(75) |
| Diuretic | 7(87.5) |
| Warfarin | 4(50) |
| ARB | 2(25) |
| Digoxin | 4(50) |
| Anti-Arrhythmic | 3(37.5) |

**Table 2. SSTR in RNAseq of myocardial tissue.** Fragments per kilobase per million mapped fragments (FPKM) corresponding to *SSTR* sequences in control and dilated cardiomyopathy heart samples analyzed by RNA sequencing. P-value by Student's T-test.

| | Control (n = 4) | Dilated cardiomyopathy (n = 10) | P-value |
|---|---|---|---|
| *SSTR1* | 0.22 | 0.30 | 0.09 |
| *SSTR2* | 0.44 | 0.53 | 0.39 |
| *SSTR3* | 0.00 | 0.01 | 0.17 |
| *SSTR4* | 0.01 | 0.01 | 0.87 |
| *SSTR5* | 2.71 | 1.53 | 0.36 |

to consider a sequence expression present in a given sample. More restrictive analyses would set the threshold at FPKM≥1. The FPKM for *SSTR* in control and dilated cardiomyopathy samples are shown in Table 2. *SSTR5* would be considered expressed in control and dilated cardiomyopathy samples by both the more and less restrictive thresholds. *SSTR1* and *SSTR2* would be considered expressed only by the less restrictive threshold, while *SSTR3* and *STTR4* would be considered not expressed in the cardiac tissue from our analysis. Based on these preliminary results, we focused on analyzing *SSTR1*, *SSTR2* and *SSTR5* in our analysis.

Gene expression, as measured by Taqman PCR, of *SSTR* in ICM myocardial samples in comparison to control hearts is shown in Fig 1A. Our results confirmed the tendency suggested by the preliminary analysis of RNAseq data. *SSTR1* tended to be upregulated in ICM tissue in comparison to control hearts, but this trend was not significant (p = 0.17 by T-test). *SSTR2* expression was unchanged in ICM tissue in comparison to control hearts. *SSTR5* was decreased in ICM samples vs. control, and this change was significant (p = 0.008). Fig 1B shows gene expression of stress and fibrosis markers in the myocardial samples from our study. Significantly increased expression of *BNP* (p = 0.04) and *MSTN* (p = 0.04) indicates ongoing cardiac stress and cardiomyocyte atrophic process characteristic of end-stage remodeling in heart failure (*10*). Unchanged levels of markers of fibrosis (*ACTA2*, *COL1A1* and *TGFβ1*) suggest a mature fibrotic scar without active pro-fibrotic gene expression.

We explored a possible correlation between altered SSTRs and markers of cardiac stress in ICM samples. We found a significant correlation between *SSTR1* expression and *BNP* (Fig 1C, p = 0.01), however, this correlation was driven by three specimens in which both *SSTR1*

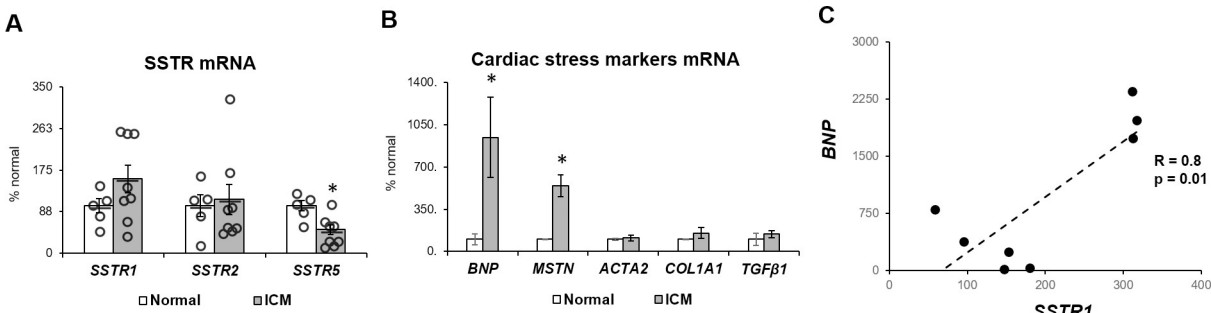

**Fig 1. Gene expression in myocardial samples.** (**A**) Gene expression of *SSTR1*, *SSTR2* and *SSTR5*, (**B**) brain natriuretic peptide (*BNP*), myostatin (*MSTN*), α-smooth muscle actin (*ACTA2*), collagen-1 (*COL1A1*), and transforming growth factor-β1 (TGFβ1) in myocardial samples from control hearts (n = 5) and ICM hearts (n = 8). Gene expression results calculated by the 2−ΔΔCT method and presented as % of the mean of control, normalized to housekeeping gene *GAPDH*. Error bars indicate standard error. *p<0.05 vs. control by T-test. Graphic representation of the correlation between *SSTRs* and *BNP* in ICM samples. (**C**) Correlation between *SSTR1* and *BNP*. White circles represent control samples and gray circles represent ICM samples.

expression and *BNP* were high, while in lower-expressing specimens this trend was not observed. There was no relationship between *SSTR5* and *BNP* or any of the other markers of cardiac stress analyzed.

## Immunostaining of SSTR

We explored the presence of SSTR in myocardial tissue by immunohistochemistry. As a positive control for specificity of the antibodies used, samples from neuroendocrine tumors and pancreatic tissue were stained concurrently. Fig 2 shows staining of SSTR1 in myocardial samples. While there was no evident positive staining of SSTR1 in control tissue, ICM tissues showed clear positive staining of SSTR1, with variability among samples (see images in last two columns). Although the signal of SSTR1 was present in all ICM specimens tested, the intensity of SSTR1 staining was comparable with the positive control neuroendocrine tumor in some specimens and stronger than the positive control in others. This is consistent with the tendency for *SSTR1* gene expression.

In the case of SSTR2, positive staining was seen in some ICM samples (Fig 3), while other specimens had negligible positive staining comparable with control hearts.

Despite SSTR5 having the highest level of the SSTR studied in terms of gene expression, staining of SSTR was low in myocardium samples in comparison with the neuroendocrine tumor positive control (Fig 4). No apparent differences were observed between control and ICM samples.

Next, we aimed to investigate colocalization between SSTR1 and SSTR2 staining and areas of fibrosis in ICM samples. Fig 5 shows that areas with positive staining for either SSTR1 or SSTR2 are often adjacent to fibrotic areas with abundant collagen (blue arrows), although not all heavily fibrotic areas showed positive SSTR staining.

As SSTR are present in certain macrophages subtypes, we explored the possibility that areas with positive SSTR staining would correspond to areas with macrophage infiltration. We did not detect positive macrophage staining in the majority of ICM samples. Fig 6A shows representative images of one ICM sample in which positive CD68 staining was found. Panels 6B and 6C highlight areas with positive CD68 macrophage staining. Although CD68 cells are

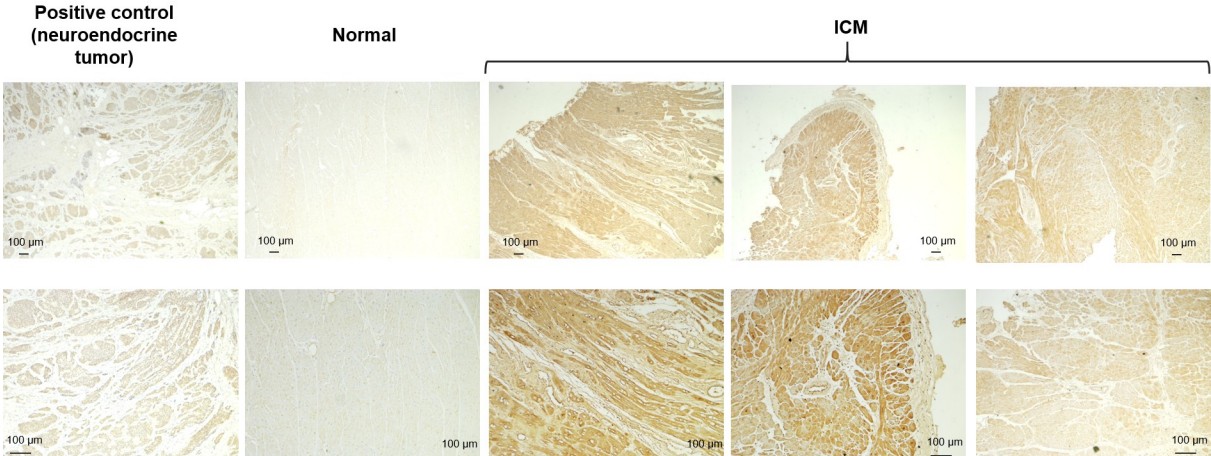

**Fig 2. SSTR1 IHC.** Representative immunohistochemistry images of SSTR1 in control and ICM myocardial samples. Staining in neuroendocrine tumor samples is used as a positive control.

**SSTR2**

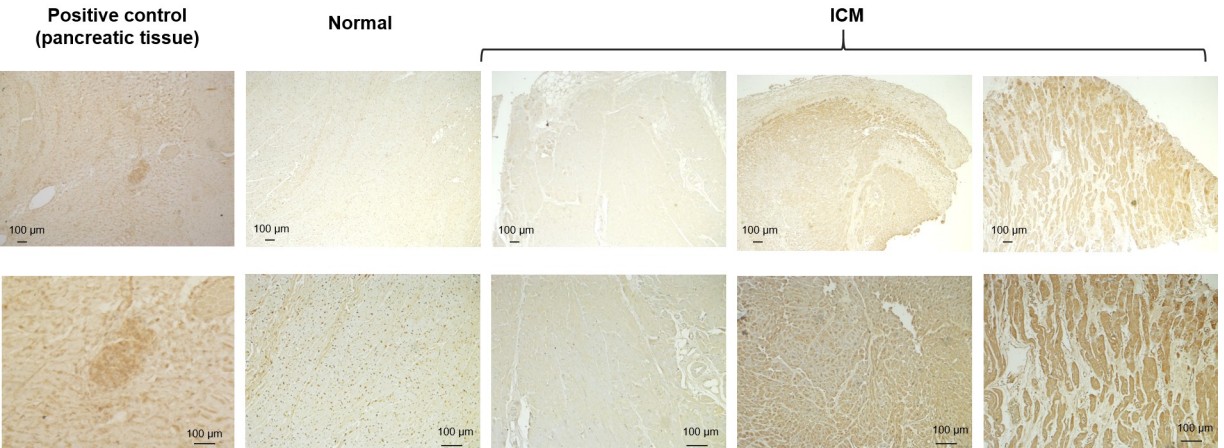

**Fig 3. SSTR2 IHC.** Representative immunohistochemistry images of SSTR2 in control and ICM myocardial samples. Staining in neuroendocrine tumor samples is used as a positive control.

positive for both SSTR1 and SSTR2 (green arrows), the majority of SSTR1 and SSTR2 positive staining localizes in fibroblast-rich areas where CD68 staining is not present (purple arrows).

## Discussion

Our results show that SSTR are occasionally upregulated in cardiac fibrotic areas not necessarily associated with increased inflammatory cell presence. Due to the variability among samples, our study does not support a possible utility of 68Ga-DOTATATE PET tracer to detect myocardial fibrosis non-invasively. However, our results raise the possibility of SSTR-rich areas in fibrotic myocardium being visible by 68Ga-DOTATATE imaging. Based on our results, when using 68Ga-DOTATATE PET tracer to detect cardiac sarcoidosis or atherosclerotic plaque, the possibility of tracer uptake in fibrotic areas should be considered.

**SSTR5**

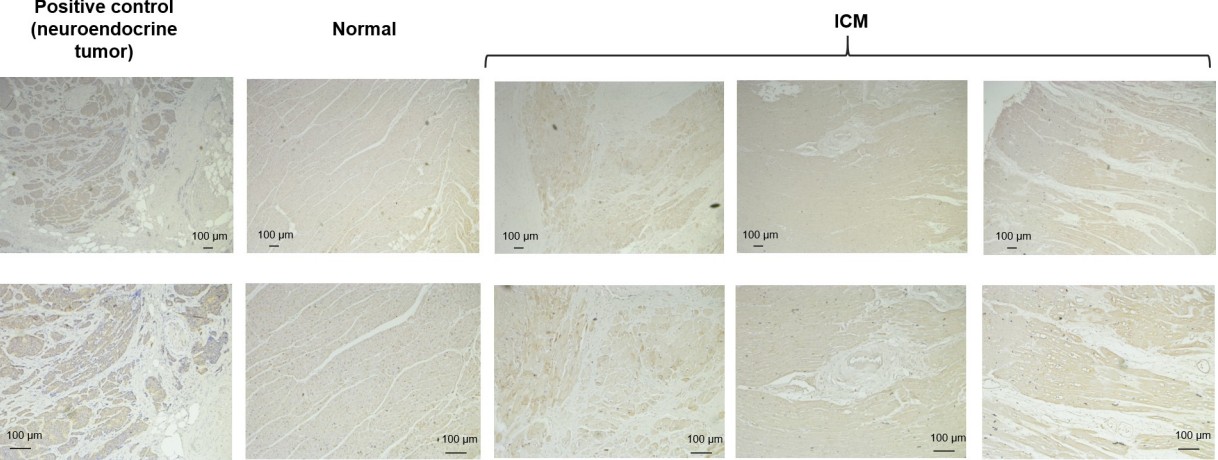

**Fig 4. SSTR5 IHC.** Representative immunohistochemistry images of SSTR5 in control and ICM myocardial samples. Staining in neuroendocrine tumor samples is used as a positive control.

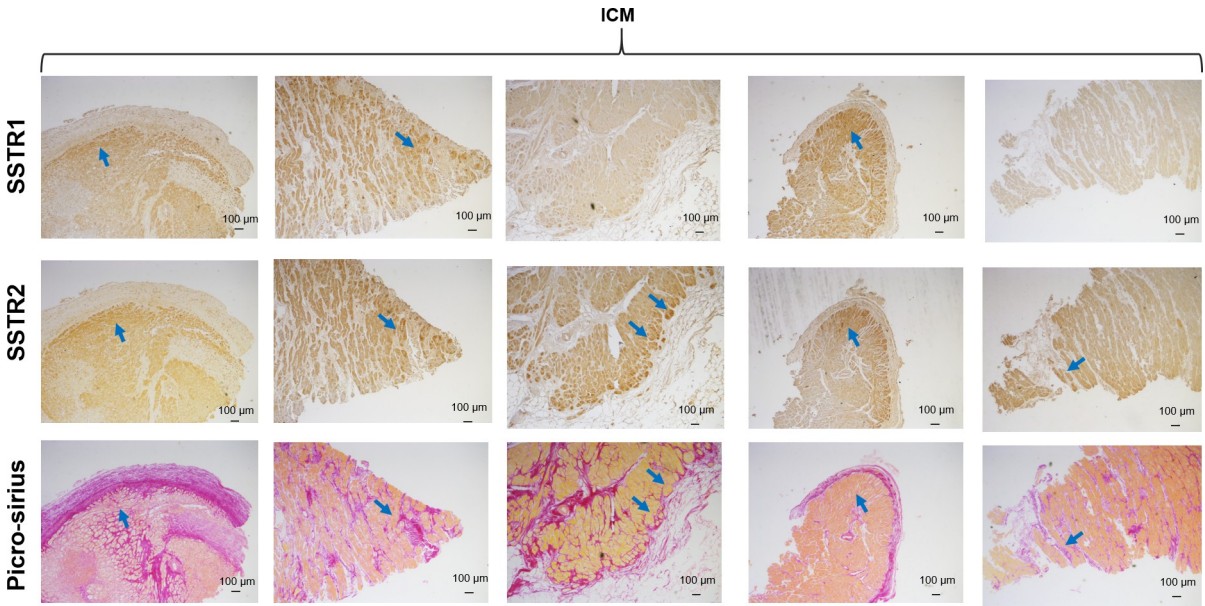

**Fig 5. SSTR IHC and fibrosis.** Immunohistochemistry images showing correspondence between SSTR1, SSTR2 and picrosirius red staining. Collagen is stained in red. Blue arrows show correspondence between areas with positive SSTR staining and fibrosis.

We showed, in a small cohort of ICM patients, that SSTR1 tends to be upregulated at the expression and protein level in end-stage cardiomyopathy myocardium with ongoing cardiac remodeling and stress and present fibrotic scar but without overly active pro-fibrotic remodeling. *SSTR2* expression was not altered in ICM patients, but its protein levels were increased. This may be consequence of postranslational processes that lead to sustained increased protein presence despite minor upregulation of gene expression. As 68Ga-DOTATATE has highest affinity for SSTR2, increased SSTR2 protein presence in fibrotic myocardium may result in some 68Ga-DOTATATE signal. Although *SSTR5* expression was significantly decreased in ICM, its low protein presence in both control and ICM myocardium suggests that its role in cardiac remodeling may not have major physiological relevance.

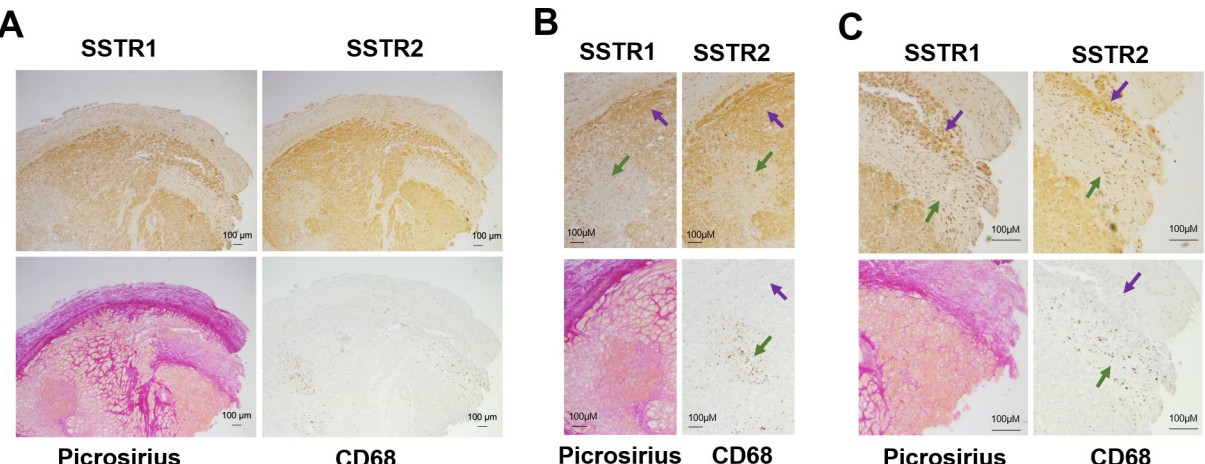

**Fig 6. SSTR IHC and macrophages.** Immunohistochemistry images showing correspondence between SSTR1, SSTR2, picrosirius red and CD68 staining. Green arrows show correspondence between areas with positive SSTR and CD68 staining. Purple arrows show areas with not correspondence between SSTR and CD68 staining.

The presence of SSTR in different cardiac cell populations has been assessed previously, with somewhat discrepant results. Bravo et. al [5] did not detect positive staining of SSTRs outside granuloma areas in hearts with sarcoidosis. Smith et al. explored the presence of SSTR in cardiac cell populations, identifying SSTR in cardiomyocytes and fibroblasts [9]. In cardiac fibroblasts, SST mobilized intracellular calcium [9]. A role for SSTR in cardiac adaptation to stress and injury has also been proposed. Vörös et al. reported cardioprotective effects of SST and SSTR1/2 in cardiac ischemia reperfusion injury [11]. SSTR1,2 and 5 were expressed in healthy human heart, with RNAscope indicating that *SSTR1,2* expression occurred in vascular endothelial cells and cardiomyocytes [11]. Interestingly, SST peptide was decreased in human ICM. Therefore, SSTR1,2 upregulation may be a compensatory response to the decrease in SST levels.

There are differences in the pattern of expression of SSTR by tissue in human versus experimental animals [12]. Therefore, results on SSTR presence in animal models should be interpreted with caution when application is sought in humans, especially with regard to as SSTR use as markers. Curtis et al. showed that *SSTR1* is the predominant subtype expressed in human blood vessels, and that the pattern of SSTR subtype expression in arteriosclerotic vessels did not differ from that found in normal vascular tissue [12]. In mice, Chen et al. showed that SSTR2 was upregulated in a model of vascular injury [13]. However, by two months after injury, the levels of SSTR2 were seen to decline [12].

As 68Ga-DOTATATE has higher affinity for SSTR2 than other SSTR, its usefulness for imaging would depend on the specific upregulation of SSTR2. For atherosclerosis, the uptake of 68Ga-DOTATATE to coronary arteries was significantly associated with the degree of plaque calcification and risk factors such as age, male sex, and hypertension [14, 15]. However, a study comparing 68Ga-DOTATATE with 68Ga-DOTANOC [16] found that 68Ga-DOTANOC is more sensitive than 68Ga-DOTATATE to detect atherosclerotic plaque in mice. Importantly, the uptake of both tracers was increased in the heart of mice with atherosclerosis [16]. 68Ga-DOTANOC has been proposed to be superior to 68Ga-DOTATATE for neuroendocrine tumor imaging due to its broader affinity to SSTR receptors (SSTR2,3,5) that detects more lesions than 68Ga-DOTATATE [17]. It is possible that a broader profile of detection is also desirable for atherosclerotic plaque detection. Activated macrophages and injured endothelium have been shown to overexpress both SSTR1 and 2 [18, 19]. A recent study has indicated that 68Ga-DOTATATE imaging may be more useful to detect large vessel vasculitis than atherosclerosis [20, 21].

In sarcoidosis, 68Ga-DOTATATE may be useful to localize granuloma-rich myocardial areas [5]. However, the signal is low and the success of granuloma visualization is conditional to low background [22]. Lee et al reported higher than ideal levels of 68Ga-DOTATATE background in patients with sarcoidosis [22]. Signal in fibrotic areas as shown in the present study may contribute to background.

Our study has several limitations. We studied a small cohort of patients with end-stage ICM who underwent cardiac transplantation or mechanical circulatory support implantation, therefore, patients will mature fibrosis but not end-stage heart failure may have different SSTR profiles which could impact 68Ga-DOTATATE PET signal. The normal cardiac specimens lacked clinical and demographic data, therefore bias related to sex or age in molecular measurements can not be ruled out. Comparable IHC staining of SSTR2 between fibrotic myocardium and the neuroendocrine tumor sample used as possible control suggests that the levels of SSTR2 in ICM samples would be sufficient to be visible by PET, however, this was not directly tested. For inflammatory cell presence, we focused on macrophages. Other inflammatory cell populations may be present in the mature fibrotic areas and contribute to SSTR signaling.

### New knowledge gained

SSTR1 and SSTR2 are occasionally upregulated in end-stage cardiac fibrotic areas without increased inflammatory cell presence. When using 68Ga-DOTATATE PET tracer to detect cardiac remodeling in the form of sarcoidosis or atherosclerotic plaque, it should be considered that fibrotic areas may be associated with unspecific signal.

## Supporting information

**S1 Data.**
(XLSX)

## Author Contributions

**Conceptualization:** Estibaliz Castillero, Giovanni Ferrari, Himanshu Gupta.

**Data curation:** Estibaliz Castillero, W. Clinton Erwin, Sameer Singh, Keith Dockery, Giovanni Ferrari, Himanshu Gupta.

**Formal analysis:** Estibaliz Castillero, W. Clinton Erwin, Sameer Singh, Nafisa Mohamoud, Elizabeth Eapen, Keith Dockery, Himanshu Gupta.

**Funding acquisition:** Giovanni Ferrari.

**Investigation:** Elizabeth Eapen.

**Methodology:** Estibaliz Castillero, Chiara Camillo, W. Clinton Erwin, Sameer Singh, Nafisa Mohamoud, Isaac George, Elizabeth Eapen, Keith Dockery.

**Project administration:** Giovanni Ferrari, Himanshu Gupta.

**Resources:** W. Clinton Erwin, Isaac George, Giovanni Ferrari.

**Software:** Himanshu Gupta.

**Supervision:** Giovanni Ferrari, Himanshu Gupta.

**Writing – original draft:** Estibaliz Castillero, Giovanni Ferrari, Himanshu Gupta.

**Writing – review & editing:** Estibaliz Castillero, Chiara Camillo, Giovanni Ferrari.

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
