## [Decision Letter · Decision Letter 0]

16 Oct 2023

PONE-D-23-27842Somatostatin receptors in fibrotic myocardiumPLOS ONE

Dear Dr. Gupta,

Thank you for submitting your manuscript to PLOS ONE. After careful consideration, we feel that it has merit but does not fully meet PLOS ONE’s publication criteria as it currently stands. Therefore, we invite you to submit a revised version of the manuscript that addresses the points raised during the review process.

We look forward to receiving your revised manuscript.

Kind regards,

Ismaheel Lawal, MD,

Academic Editor

PLOS ONE

 [This work was supported, in part, by a National Heart, Lung and Blood Institute of the National Institutes of Health (R01-HL131872) (to G.F). S.S. received funding from the Thoracic Surgery Foundation Resident Research Award ].  

3. We note that Figure 1, 3, 4, 5, 6, 7, S1 and S2 in your submission contain copyrighted images. All PLOS content is published under the Creative Commons Attribution License (CC BY 4.0), which means that the manuscript, images, and Supporting Information files will be freely available online, and any third party is permitted to access, download, copy, distribute, and use these materials in any way, even commercially, with proper attribution. For more information, see our copyright guidelines: http://journals.plos.org/plosone/s/licenses-and-copyright.

A. You may seek permission from the original copyright holder of Figure 1, 3, 4, 5, 6, 7, S1 and S2 to publish the content specifically under the CC BY 4.0 license. 

B. If you are unable to obtain permission from the original copyright holder to publish these figures under the CC BY 4.0 license or if the copyright holder’s requirements are incompatible with the CC BY 4.0 license, please either i) remove the figure or ii) supply a replacement figure that complies with the CC BY 4.0 license. Please check copyright information on all replacement figures and update the figure caption with source information. If applicable, please specify in the figure caption text when a figure is similar but not identical to the original image and is therefore for illustrative purposes only.

Reviewers' comments:

Reviewer's Responses to Questions

**Comments to the Author**

1. Is the manuscript technically sound, and do the data support the conclusions?

Reviewer #1: Yes

Reviewer #2: Yes

2. Has the statistical analysis been performed appropriately and rigorously? 

Reviewer #1: Yes

Reviewer #2: I Don't Know

3. Have the authors made all data underlying the findings in their manuscript fully available?

Reviewer #1: Yes

Reviewer #2: Yes

4. Is the manuscript presented in an intelligible fashion and written in standard English?

Reviewer #1: Yes

Reviewer #2: Yes

5. Review Comments to the Author

Reviewer #1: Somatostatin receptor is expressed in sarcoid granulomas, and preliminary clinical studies have shown that myocardial sarcoidosis can be identified on somatostatin receptor (SSTR)-targeted PET and 68Ga-DOTATATE PET/CT for diagnosis and response assessment in cardiac sarcoidosis is clinical tools. The present study was aimed to test whether SSTR could be upregulated in cardiac fibrotic scar, and found that using IHC and gene analysis STR1 and SSTR2 are occasionally upregulated in end-stage cardiac fibrotic areas without increased inflammatory cell presence. The manuscript is well written, although parts of introduction and discussion should be more focused and structured, and the patients data are very limited.

Major concerns:

1) In order to compare the relative expression of SSTR and make unbiased comparison, the controls group demographic data have to be presented as well. The SSTR may change due to sex and age, please discuss that.

2) Only 50% of HF patients have previous MI, do they have similar magnitude in changes of SSTR in compared to patients with no previous MI. Do LVEF and SSTR show positive correlation?

3) Case report - its not really fit into the global scope of the MS, the patient had a relative fresh MI, scar/healing process is completed? In addition, inflammation was detected? Similar Suppl Fig 2 is unnecessary to show.

4) It will be ideally and more supportive if the authors would provide 68Ga-DOTATATE PET/CT data; including IM cardiomyopathy patients. The n numbers are extremely low, this is not enough for proper conclusion and interpretations.

5) mRNA expression levels, SSTR show no significant changes between the group, but was a large differene in protein expression levels, please discuss that.

6) Figure 4 - SSTR2 seems to be expressed in cardiomyocytes as well, however the authors stated that "We showed, in a small cohort of ICM patients, that SSTR1 tends to be upregulated at the expression and protein level in end-stage fibrotic myocardium with ongoing cardiac remodeling and stress but without overly active pro-fibrotic remodeling". In order to clarify that the investigators should also apply IHC staining for cardiomyocytes.

Reviewer #2: The authors performed a hard work. I assume that to recruit 8 suitable patients they had to analyze a large number of patients, with a large dropout rate, which is indirectly evidenced by the time interval for the selection of 11 years (from 2010 to 2021).

A unique material was obtained - tissue samples of healthy hearts from donors (coincidentally, turned out to be unclaimed), tissue samples of hearts of patients with terminal CHF who underwent heart transplantation or implantation of left ventricular mechanical support devices.

The study of heart tissue samples from these patients by PCR and determination of somatostatin receptor (SSTR1, SSTR2, SSTR3, SSTR4, SSTR5) mRNA concentrations was performed.

In the introduction and discussion, the authors refer to the latest most significant works in this field, including their previous works on this topic, the authors are well familiar with the research topic and are deeply immersed in the problems of the discussed area. The analysis of the premise of this paper is well done.

But when analyzing the work, I have questions for the IHC block, which need to be clarified if we want to be sure of the accuracy of the results obtained and their correct interpretation.

For a more correct interpretation and to determine the quality of the IHC staining performed and to ensure the reproducibility of the results, the manuscript lacks information about the IHC technique itself, namely the IHC protocol, what buffer solution was used, the imaging system used and the secondary antibodies (avidin-biotin imaging method or other, horseradish peroxidase or alkaline phosphatase, what reagents were used and how long the peroxidase block was performed).

There are no questions to the interpretation of Figures 2-5, in my opinion, the authors have evaluated and described them correctly.

In Figure 7, the presented SSTR1, SSTR2 preparations show intense positive staining of cardiomyocytes (CMC), weak staining of fibrosis areas, and intense staining of individual cells in the fibrosis area (presumably macrophages).

Based on previously published works (5) (https://pubmed.ncbi.nlm.nih.gov/31197742/ ), it was shown that the positive expression of SSTR1 and SSTR2 by activated macrophages in the area of sarcoid granulomas is well detected by IHC, while the rest of the heart tissue, in particular, CMC are negative in staining. Positive staining of macrophages by IHC is expected. Intense staining of SMCs is an unexpected result that requires additional verification and caution in interpreting the results.

Nonspecific staining should be ruled out for a more reliable interpretation of the results.

One possible reason for nonspecific staining of cardiomyocytes is the use of polyclonal antibodies; the presence of antibodies to multiple epitopes may increase the likelihood of cross-reactivity with other proteins. There is less risk of cross-reactivity when monoclonal antibodies are used.

If possible, if paraffin blocks with tissue samples are preserved, it makes sense to make additional slices from the same blocks and stain them with monoclonal antibodies to SSTR1, SSTR2.

Neuroendocrine tumor tissue (NEO) was used as a positive control for IHC. The authors' decision to use NEO as a control is understandable; this tumor can serve as a link between two methods - PET and IHC, if in the same patient this tumor demonstrated 68Ga-DOTATATE accumulation in PET in vivo, and the same tumor after tumor removal and biopsy demonstrates positive staining for somatostatin receptors in IHC.

However, NEO may have angstigenic heterogeneity, in combination with polyclonal antibodies, this may reduce the reliability of the result obtained.

At least for SSTR2, it makes sense to use pancreatic tissue as an additional control (positive - islets of Langerhans, and negative - the rest of the pancreatic tissue).

Another possible reason for nonspecific staining of cardiomyocytes when using the immunoperoxidase staining system is insufficient inhibition of endogenous peroxidases. All samples containing hemoproteins, including myoglobin in muscle cells, hemoglobin in erythrocytes, and cytochrome in granulocytes and monocytes, may show unsuppressed endogenous peroxidase activity. There is experimental evidence that myoglobin concentration in cardiomyocytes may increase during prolonged hypoxia (https://pubmed.ncbi.nlm.nih.gov/19005161/ ; https://pubmed.ncbi.nlm.nih.gov/20675543/ ).

Intense positive staining of cardiomyocytes appeared in patients with severe heart failure, it is known that tissues of such patients suffer from hypoperfusion and experience hypoxia, which, theoretically, can lead to increased expression of myoglobin and increased activity of endogenous peroxidases. To eliminate the influence of this factor in IHC staining, change the peroxidase block or increase the tissue incubation time with it or choose a different tagging enzyme such as alkaline phosphatase.

The above factors may change the answer to the main question: What exactly is the tissue substrate for PET imaging when using somatostatin-directed radiopharmaceuticals (68Ga-DOTATATE)? Previous work suggests that this substrate is very likely to be inflammatory cells, particularly macrophages. This work claims to identify a fundamentally new substrate - directly the SMCs themselves, under certain conditions (terminal CHF). Since this work claims to validate fundamentally new knowledge, this requires additional careful scrutiny to avoid unpleasant mistakes. This is the basis for my recommendations.

I believe that the paper can be considered for publication after the above points have been elaborated or clarified.

Minor issues requiring clarification/correction:

1) If the mean mRNA content of SSTR1 and SSTR2 did not differ between the study group and the control group, how to explain this difference with respect to the IHC pattern, where differences in staining intensity are clearly visible?

2) A clinical case of a patient with a combination of NEO and ischemic cardiopathy is given as background. Unfortunately, I could not understand from the text of the article whether any of the patients in the main group (n=8, with ischemic cardiopathy) was diagnosed with HEO, or whether all patients in the main group were without HEO? I think this point is important; it may be better to spell it out more clearly. If the patients were without NEO, what were the indications for PET with 68Ga-DOTATATE?

3) Numbering of figures is confused - Figure 1 in the text corresponds to Figure 2 in the appendix, Figure 2 in the text corresponds to Figure 3 in the appendix, etc.

6. PLOS authors have the option to publish the peer review history of their article (what does this mean?). If published, this will include your full peer review and any attached files.

Reviewer #1: No

Reviewer #2: No

---

## [Author Response · Author response to Decision Letter 0]

13 May 2024

All questions and concerns have been addressed in the revised manuscript. Response to reviewers' contains our detailed responses.

---

## [Editor Report · Decision Letter 1]

20 May 2024

Somatostatin receptors in fibrotic myocardium

PONE-D-23-27842R1

Dear Dr. Gupta,

We’re pleased to inform you that your manuscript has been judged scientifically suitable for publication and will be formally accepted for publication once it meets all outstanding technical requirements.

Kind regards,

Ismaheel Lawal, MD,

Academic Editor

PLOS ONE
---

## [Editor Report · Acceptance letter]

10 Jul 2024

PONE-D-23-27842R1 

PLOS ONE

Dear Dr. Gupta, 

I'm pleased to inform you that your manuscript has been deemed suitable for publication in PLOS ONE. Congratulations! Your manuscript is now being handed over to our production team.

Kind regards, 

on behalf of

Dr. Ismaheel Lawal 

Academic Editor

PLOS ONE